# Immuno-Haematologic Aspects of Dengue Infection: Biologic Insights and Clinical Implications

**DOI:** 10.3390/v16071090

**Published:** 2024-07-06

**Authors:** Tan Jiao Jie Cherie, Clarice Shi Hui Choong, Muhammad Bilal Abid, Matthew W. Weber, Eng Soo Yap, Suranjith L. Seneviratne, Visula Abeysuriya, Sanjay de Mel

**Affiliations:** 1Department of Medicine, National University Health System, Singapore 119228, Singapore; tanjiaojie@gmail.com; 2Department of Haematology Oncology, National University Cancer Institute, National University Health System, Singapore 119228, Singapore; clarice_choong@nuhs.edu.sg; 3Division of Haematology/Oncology, Department of Medicine, Medical College of Wisconsin, Milwaukee, WI 53226, USA; mabid@mcw.edu (M.B.A.); mwweber@mcw.edu (M.W.W.); 4Department of Laboratory Medicine, National University Health System, Singapore 119228, Singapore; eng_soo_yap@nuhs.edu.sg; 5Institute of Immunity and Transplantation, Royal Free Hospital and University College London, London NW3 2PP, UK; 6Nawaloka Hospital Research and Educational Foundation, Nawaloka Hospitals PLC, Colombo 00200, Sri Lanka; 7Department of Immunology, Institute of Biochemistry, Molecular Biology and Biotechnology, University of Colombo, Colombo 00300, Sri Lanka; visulasrilanka@hotmail.com

**Keywords:** dengue, immuno-haematology, thrombocytopaenia, immune dysregulation, haemostasis

## Abstract

Dengue infection is caused by the dengue virus (DENV) and is transmitted to humans by infected female *Aedes aegypti* and *Aedes albopictus* mosquitoes. There are nearly 100 million new dengue cases yearly in more than 120 countries, with a five-fold increase in incidence over the past four decades. While many patients experience a mild illness, a subset suffer from severe disease, which can be fatal. Dysregulated immune responses are central to the pathogenesis of dengue, and haematologic manifestations are a prominent feature of severe disease. While thrombocytopaenia and coagulopathy are major causes of bleeding in severe dengue, leucocyte abnormalities are emerging as important markers of prognosis. In this review, we provide our perspective on the clinical aspects and pathophysiology of haematologic manifestations in dengue. We also discuss the key gaps in our current practice and areas to be addressed by future research.

## 1. Introduction

Dengue infection is caused by the dengue virus (DENV), a positive-sense, single-stranded RNA virus, which belongs to the Flaviviridae family and is transmitted by *Aedes aegypti* and *Aedes albopictus* mosquitoes [1]. DENVs comprise four serotypes, which are distinguished based on differences in viral structural and non-structural proteins [2]. Infection with each serotype confers lifelong immunity for the causative serotype but not for others [3]. The incidence of dengue is on the increase, with 100–400 million new infections each year and the highest prevalence observed in countries with tropical climates [4]. It is noteworthy that only a quarter of these patients become symptomatic [5]. Clinical manifestations may be limited to mild fever, headache and myalgia, or they may progress to severe dengue. 

The 2009 World Health Organization (WHO) criteria classify dengue according to three levels of severity, including dengue with and without warning signs and severe dengue, with the aim of guiding management [6]. Dengue infection is most commonly diagnosed using the enzyme-linked immunosorbent assay (ELISA), which measures dengue immunoglobulin G (IgG) and immunoglobulin M (IgM) antibodies [7]. The alternatives include the non-structural antigen 1 (NS1) or the dengue reverse transcriptase PCR (RT-PCR) assay [8]. The management of dengue mainly comprises supportive therapy, which is adequate for most patients [9]. However, a subset of patients develop severe disease for which optimal management remains uncertain. 

Severe dengue occurs in 500,000 people per year, with a mortality rate of 10% among hospitalised patients [10]. A dysregulated host immune response, including antibody-dependent enhancement (ADE) [11], is thought to drive most of the complications associated with severe dengue, in particular the haematologic manifestations [12]. Haematologic abnormalities are a key feature of dengue, with coagulopathy (Figure 1), thrombocytopaenia (Figure 2) and leucocyte abnormalities (Figure 3) being well described. The pathogenesis and management of these are the subject of active research. In this review, we provide our perspective on the immuno-haematological aspects of dengue infection with a focus on their biological and clinical implications. 

## 2. Haemostatic Abnormalities in Dengue Infection

Haemostatic dysfunction resulting in derangement of laboratory coagulation assays is relatively common in dengue infection (Figure 1). A meta-analysis of forty-two studies reported the frequency of prolonged activated partial thromboplastin time (aPTT) and prothrombin time (PT) among patients with dengue at 42.9% and 16.5%, respectively [13]. Despite the high frequency of prolonged APTT and PT, clinically significant bleeding manifestations are less common [14]. In the event of bleeding, blood transfusion requirements are low. In a retrospective study conducted over 7 years in a tertiary hospital in south India, clinically significant bleeding manifestations occurred in 44.7% of patients with dengue infection, but only 1.2% of them required packed cell transfusion [15].

The specific mechanism underlying coagulation abnormalities associated with dengue remains unclear. Various pathogenic mechanisms have been proposed. The aPTT and PT serve as indicators for the intrinsic and extrinsic pathways of the coagulation system, respectively [16]. Prolongation of PT and aPTT may result from either decreased synthesis of specific factors or increased consumption of these factors [16]. The NS1 protein can bind to both thrombin and prothrombin [17]. While binding to thrombin has no effects, it inhibits the activation of prothrombin [18]. NS1 also mediates the activation of plasminogen via plasminogen cross-reactive antibodies, thereby enhancing fibrinolytic activity and bleeding [19]. DENV-induced hepatocellular injury also contributes to coagulopathy via reduced synthesis of coagulation factors and may at least partly explain the prolongation of both aPTT and PT [20,21].

An imbalance between coagulation and fibrinolysis is also postulated to cause haemorrhagic complications in DHS and DSS [22]. DENV-induced macrophage migration inhibitory factor promotes bleeding by inducing platelet-activating factor [23]. Furthermore, increased IL-6 downregulates the production of factor XII, which initiates the intrinsic coagulation pathway, resulting in prolongation of the aPTT [24]. Excessive tissue plasminogen activator (tPA) production in DHS/DSS leads to hyperfibrinolysis and further exacerbates the bleeding tendency [25].

Indeed, dengue patients with bleeding complications were shown to have higher levels of tPA, D-dimer and reduced thrombin formation compared to those without bleeding [26]. Similar results were demonstrated in children with DHF, where von Willebrand factor antigen (vWF: Ag), tissue factor (TF) and plasminogen activator inhibitor (PAI-1) were increased, but ADAMTS-13 (a dis-integrin and metalloprotease with thrombospondin repeats) was significantly reduced [27]. It is likely that the rises in vWF: Ag, TF and PAI-1 are a response to the acute inflammatory state in DHF [28], and the concomitant rise in tPA and consumption of coagulation factors tips the balance in favour of bleeding. Furthermore, excessive vWF may lead to abnormal platelet activation and aggregation, contributing to thrombocytopaenia and bleeding [24,28].

Disseminated intravascular coagulation (DIC) is also a major contributor to dengue-related coagulopathy and was recently reported to occur in 26% of patients with severe disease, in addition to being a predictive factor for early mortality [29]. Taken together, these data suggest that reduced thrombin formation and increased fibrinolytic activity are the key contributing factors to bleeding complications in dengue infection. It is noteworthy that dengue-related coagulopathy is a result of both the direct effects of the virus on the haemostatic system as well as the dysregulated host immune response [23].

There is currently a paucity of predictive factors to predict bleeding manifestations due to dengue. Further studies are required to identify patients who may need closer monitoring of their coagulation system and potentially earlier intervention. Furthermore, the optimal management of dengue-related coagulopathy needs to be established, especially with regards to the indication and timing for the use of blood products. 

## 3. Thrombocytopaenia 

Thrombocytopaenia is a major contributor to the haemorrhagic tendency in dengue infection, occurring in up to 79% of cases, with the elderly being at higher risk [30]. Dengue-related thrombocytopaenia typically occurs on days 4–6 and recovers by day 10 [31]. The platelet nadir is variable, ranging from 40 to 90 × 10^9^/L with a median nadir of 65 × 10^9^/L according to a prospective observational study conducted among 2300 children with dengue infection [32].

In the acute phase of secondary dengue, the DENV binds to platelets, forming platelet-associated IgM (PA-IgM), resulting in platelet destruction [33]. In young patients with DHF, platelet aggregation is decreased, and the plasma levels of platelet factor-4 and beta thromboglobulin are increased, contributing to bleeding complications [34]. Interleukin-1B, released from platelet microparticles via inflammasome activation, also promotes vascular permeability, leading to plasma leakage [35]. It is noteworthy that the levels of platelet microparticles and plasma P-selectin correlate with the onset and recovery from thrombocytopaenia in dengue infection [36]. In addition, complement-C5b-9-mediated apoptosis of activated platelets and clearance of IgG-bound platelets have been observed to occur on days 4–6 of dengue infection, accounting for the significant drop in platelet count from day 4 onwards [31]. The consumption of platelets is also accelerated by DENV-induced disseminated intravascular coagulation (DIC), as discussed above. 

DENV-induced bone marrow suppression is proposed to occur within 3–4 days of dengue infection [37]. The reduction in haematopoiesis is likely a protective mechanism to limit the injury to the bone marrow progenitor cells during elimination of dengue-infected cells [38]. Bone marrow suppression is likely to contribute significantly to DENV-related thrombocytopaenia. The fact that the immature platelet fraction (a surrogate for thrombopoiesis) in the peripheral blood is significantly lower in patients with severe dengue supports this hypothesis [39]. This is, however, difficult to study definitively, as bone marrow aspiration/biopsy is rarely performed in patients with dengue infection. A variety of clinical parameters have been proposed, which may enable early identification of patients who are likely to develop severe thrombocytopaenia [39,40,41,42]. Further studies are required to validate these findings and derive globally applicable prognostic scores for dengue-related thrombocytopaenia.

## 4. Leucocyte Abnormalities

### 4.1. Monocytes and Macrophages

Monocytes are a key target of the DENV, and, while being crucial for the host anti-viral response, they also contribute to ADE [43]. Several monocyte subsets have been implicated in the pathogenesis of dengue [44], and the DENV appears to have a predilection to infect some subsets over others. The biological role of specific monocyte subsets in dengue remains an active field of research [45,46]. CD14+/CD16+ intermediate monocytes were found to be increased in paediatric patients with dengue up until day 3 of infection, followed by a decline from day 4 onwards [47]. Interestingly, intracellular staining demonstrated DENV infection in the classical (CD14 bright/CD16−) and intermediate monocyte subsets but not in the non-classical monocytes. The role of monocytes was reinforced in this study through the detection of raised IL-10 mRNA levels from CD14+ monocytes compared to other subsets of peripheral blood mononuclear cells. The increased IL-10 levels resulted in upregulation of pathways inhibiting the anti-viral response, including the suppression of JAK-STAT signalling [48]. Furthermore, in vitro infection of monocytes with the DENV resulted in suppression of Interferon-Beta and induced nitric oxide synthesis with sub-neutralising concentrations of DENV-specific antibodies [12]. Collectively, these data suggest that the DENV “hijacks” monocytes to downregulate innate anti-viral immune responses. 

DENV-infected macrophages have been detected in the tissues of patients with dengue [49], and macrophage recruitment to the endothelium has been implicated in the development of haemorrhage [50]. It has been postulated that macrophages at different anatomic sites may have distinctive roles in the response to dengue infection, with some subsets being permissive to DENV [44]. Splenic macrophages in particular were shown to be permissive to DENV infection compared to B cells and may play a crucial role in immune enhancement and ADE [51,52]. Given this intriguing finding, future studies should characterise splenic macrophages in patients with severe dengue with the aim of unravelling how they differ phenotypically from patients with non-severe disease. 

It is also plausible that macrophage dysfunction underlies dengue-related haemophagocytic lymphohistiocytosis (HLH). HLH is characterised by inappropriate macrophage stimulation, resulting in a cytokine storm and a severe systemic inflammatory response [53]. Variable levels of haemophagocytosis were seen in post-mortem studies of patients with severe dengue, in contrast to those with mild disease [54]. In a retrospective single-centre study, mortality due to severe dengue was 22%, and approximately 40% of these patients had dengue-related HLH [53]. The phenotype of macrophages in dengue-associated HLH has not been adequately characterised and is an important area to be addressed in future research. 

Further studies are required to further unravel the biological significance of the monocyte/macrophage lineage, especially in the pathogenesis of severe dengue. The integration of single-cell RNA sequencing with flow cytometry at serial time points of the disease would provide crucial information on the evolution of monocyte subsets and their function. Differential gene expression analysis between monocyte subsets infected with DENV and those not infected would also be of significant interest. Ideally, the changes in monocyte subsets should be studied in the context of other immune cell populations, including lymphocytes, as it is likely that a complex interplay exists between these populations [55].

### 4.2. Lymphocytes

Lymphocytes play a crucial role in the immune response against viral infections, including dengue [56]. Interestingly, a lower mean lymphocyte count was reported as a biomarker to distinguish dengue from non-dengue viral fevers [57], while a higher percentage of lymphocytes in the WBC differential was associated with a shorter hospital stay in a study from India [58]. A lower absolute lymphocyte count was also reported as a predictor of severe dengue in a separate study [57]. These conflicting data suggest that specific subsets of lymphocytes (rather than total lymphocytes) may be more relevant in the pathophysiology of dengue infection.

The finding of “atypical” or “reactive” lymphocytes in dengue infection has been well described for several years [59]. Atypical lymphocytes (ALs) are readily detected by evaluation of the peripheral blood film [60], and more recently, by automated haematology analysers [61]. The current analysers identify ALs based on their light scatter properties, which generates the AL count (a research parameter) as part of the full blood count readout. These data have been used by several groups to describe the kinetics of ALs and their potential prognostic significance [62]. ALs can be detected at presentation, are known to peak during the defervescing phase of dengue and are associated with more severe disease [60]. Specifically, the percentage of AL at presentation was associated with more severe thrombocytopaenia [41,60], as well as haemorrhage, shock and respiratory compromise [41]. It is reasonable to postulate that a higher AL count reflects greater immune dysregulation, resulting in more severe thrombocytopaenia and other clinical manifestations.

T cells play a significant role in the host response to dengue infection. Patients with severe dengue were shown to have an increase in the markers of T-cell activation, as well as cytolytic granule proteins [63]. In contrast, other studies have shown that the inhibitory CTLA4 antigen was expressed at higher levels in T cells from patients with dengue, suggesting that T-cell responses against DENV may be attenuated in some patients [64]. It is noteworthy that T-regulatory (T-reg) cells are increased in dengue infection; however, they are not associated with adverse clinical outcomes and are of the naïve T-reg subtype, with limited suppressive capacity [65]. These data suggest a heterogeneity of T-cell subsets in dengue, which may be evolving as the disease progresses. Single-cell RNA sequencing (scRNAseq) at serial time points during dengue infection demonstrated unique effector T-cell clusters expressing skin homing signature genes on day 1 of illness, which was associated with defervescence [66]. Further studies using high-dimensional profiling of T cells in dengue are required to elucidate the key T-cell subsets involved in the progression to severe disease. Prospective studies looking at the correlation between T-cell subsets and clinical outcomes for DENV would be of significant interest and may generate new hypotheses to be addressed through translational research.

While B cells are important in generating humoral response against viral infections, they can also be a target for infection by the DENV [67]. Indeed, virus-inclusive single-cell RNA-Seq (viscRNA-Seq) analysis demonstrated that naïve B cells contained the highest amount of DENV RNA among the immune cells analysed in patients with severe dengue [68]. The DENV has been proposed to bind to B cells via CD300a and to be internalised via clathrin-mediated endocytosis [69]. Infection of B cells by the DENV was shown to result in their proliferation as well as differentiation to a plasmablastic phenotype [70]. This process of B-cell proliferation and differentiation is likely to be important in the generation of a robust humoral immune response against the virus [67]. In vitro studies have confirmed that direct stimulation by the DENV results in secretion of IgM and IL-6 by the host B cells [70]. Furthermore, the identification of a subset of CD24/CD38-double-positive, CD27-negative B cells, which did not produce TNFα or IL-10 upon in vitro stimulation, was associated with severe dengue in a paediatric study from Cambodia [71]. Given their “lymphoplasmacytic” morphology, it is reasonable to speculate that the ALs detected in peripheral blood films of DENV patients may be activated B cells. The finding of a dominant CD19-expressing lymphoid population on peripheral blood flow cytometry in DF supports this hypothesis [72]. Taken together, these data support the important role of B-cell activation in the host response to dengue infection.

It is also noteworthy that the antibody response to the DENV is implicated in the pathogenesis of severe disease [11]. Indeed, skin and gut homing genes were upregulated in the plasma cells and plasmablasts of dengue patients during the febrile period [66], which may be a signature of entry into the critical phase. ADE has been proposed as the key mechanism behind the higher frequency of severe dengue in secondary infections [71]. The antibodies produced by B cells in primary dengue infection may be ineffective at neutralising the virus in subsequent infections [73]. Indeed, the immune complexes generated by pre-existing antibodies promote entry of the virus into monocytes and macrophages via FCγ receptors, leading to more severe disease [71,74]. Notably, the Fcγ receptor FcγRIIB and the Fc-like receptor LILRB1 expressed by B cells are upregulated in dengue infection and have been implicated in ADE [71]. The interplay between B-cell activation, the humoral immune system and macrophage subsets in the pathophysiology of ADE is an important question to be addressed in future studies. It is also possible that the ALs detected by automated haematology analysers represent a subset of B cells, which have been infected by DENV. This may explain their association with more severe disease. Future studies should focus on delineating the phenotype of DENV-infected B cells in clinical samples at serial time points to confirm this hypothesis.

Natural killer (NK) cells are a key part of the innate immune response against viral infections and are characterised by their cytotoxic function [75]. NK cells secrete cytolytic granules to eliminate dengue-infected cells [76]. The significance of NK cells in the immune response against dengue has since been corroborated in clinical studies. The CD56 bright immunoregulatory subset of NK cells was found to be increased in patients with mild disease but notably not in those with severe dengue [77]. Similarly, the cytotoxic CD56 dim NK cell subset showed increased cytolytic capability (based on the expression of CD69, NKP30,Granzyme B and IFNγ) in those with mild compared to severe dengue [71]. Importantly, the impaired cytotoxic capacity of NK cells in those with severe disease persisted even up to a week after infection [71]. Mass cytometry via time of flight (CyTOF) analysis was able to delineate unique NK cell signatures associated with dengue infection in paediatric and adult patients, respectively [78]. Similar studies evaluating NK cell subsets via high-dimensional immune profiling at presentation and at serial time points during the disease would be of significant value, especially if NK cell phenotypes at presentation can predict the severity of disease. The impact of dengue vaccines on T/NK cell subsets and their function is another important area to be addressed in future research. We hypothesise that the key lymphoid subsets driving severe dengue are activated B cells as well as dysregulated effector T/NK cells.

### 4.3. Neutrophils

Neutropenia is a well-recognised feature of many viral infections, including dengue [79]. The neutrophil nadir in dengue infection typically occurs on day 4–6 and may coincide with defervescence [80]. It is noteworthy that severe neutropenia is not associated with disease severity or mortality [78]. Although neutropenia is common in dengue, there is clear evidence of neutrophil activation during dengue infection, as evidenced by elevated levels of IL-8 and TNF alpha [81,82].

Neutrophils in dengue also overexpress the activation marker CD66B, which potentiates endothelial adhesion and the generation of reactive oxygen species [83]. Patients with severe dengue also have increased neutrophil elastase levels, which implies that enhanced neutrophil activation can also be associated with severe disease. The generation of neutrophil extracellular traps (NETs) (web-like chromatin structures, which trap and destroy pathogens) has recently been demonstrated in dengue [83]. In addition to their antimicrobial functions, NETs are also proposed to have pro-inflammatory properties and may exacerbate vascular permeability. Indeed, increased levels of NET components were found in the serum of patients with severe dengue.

The occurrence of NETs in dengue is thought to be mediated by platelet extracellular vesicles, which in turn promote signalling via C-type lectin domain containing 5 A (CLEC5A) and Toll-like receptor 2 (TLR2) [84]. Collectively, these data support a prominent role for neutrophils in generating the pro-inflammatory milieu associated with severe dengue. Future studies should address the neutrophil subsets involved in this process and how they interact with other immune cells, as well as virus-infected cells, in driving the pathogenesis of severe dengue.

### 4.4. Mast Cells

Mast cells are a key component of the host response against parasitic infection [85]; however, their role in viral infections remains less clear. Mast cell activation, as measured by mast-cell-derived mediators, such as vascular endothelial growth factor (VEGF), tryptase and chymase, has been implicated in the pathogenesis of dengue infection. However, the evidence related to the role of mast cell activation is mixed, both in mild and severe dengue [86]. Although elevated levels of tryptase are associated with severe dengue [85], further research is required to determine the true clinical relevance of mast cells in dengue infection.

### 4.5. Eosinophils and Basophils

The role of eosinophils and basophils in the pathophysiology of dengue infection remains relatively understudied. Low eosinophil counts seem to occur in the acute phase of dengue, while eosinophilia has been reported later in the disease, with a peak on days 9–10 [87]. There have been no studies looking at whether eosinophil or basophil counts correlate with the severity of dengue. An important confounding factor to be considered in future studies is the possibility of concurrent parasitic infection (which can also cause eosinophilia or basophilia), which may be prevalent in the same populations susceptible to dengue.

## 5. Implications for Prognosis and Risk Stratification

Early identification of patients at risk of severe dengue infection is crucial to enable patients at risk to be offered closer monitoring and appropriate investigations. Given that dengue occurs predominantly in resource-limited settings, this would also allow the prioritisation of scarce healthcare resources for those patients who need them most. The risk stratification for dengue is currently based on clinical parameters, as described in the WHO guidelines on dengue [40]. It is also noteworthy that age and co-morbidities may confer an increased risk of severe morbidity from dengue—a key factor to consider, given the ageing populations around the world [88]. In a retrospective case–control study conducted in Singapore, dengue patients aged above 60 years had a 2.75 times higher risk of severe dengue compared to those aged 12–29 years [89]. Similarly, patients with pre-existing co-morbidities had a 1.63 times higher risk of severe dengue than patients without co-morbidities, with diabetes and cardiac disorders being of particular relevance [89]. While these data are a valuable guide to the risk stratification of patients, clinical risk stratification alone often lacks specificity. It is noteworthy that there are currently no validated laboratory investigations, which can predict the severity of dengue infection [61]. There are, however, emerging data on haematologic and immunologic parameters, which have potential utility in this space [89].

A raised haematocrit has been proposed as a marker of haemoconcentration, and hence, a surrogate for vascular leakage in dengue infection. The WHO guideline on dengue incorporates a rising haematocrit of more than 20% as a marker of severity [90]. Indeed, it is noteworthy that a haematocrit value greater than 40% has been associated with clinical capillary leak syndrome [91]. Other haematologic parameters, which may have utility, include the AL count, which is predictive of thrombocytopaenia as well as other adverse clinical outcomes [83]. Accurately predicting the severity of thrombocytopaenia, as well as the timing of its recovery, is also of great clinical value and has been the subject of several studies [39,41,61]. The immature platelet fraction (IPF) is an index of thrombopoiesis, which quantifies reticulated platelets that have recently been released from the bone marrow [92]. An IPF of more than 10% after defervescence predicts subsequent platelet recovery to a haemostatic level of more than 60 × 10^9^/L within 72 h [93]. In a prospective study conducted among 240 patients, an IPF greater than 7.25% on day 3 of illness had a sensitivity of 88% for predicting platelet recovery to >60 × 10^9^/L on day 8 of illness. Within 48 h after the peak IPF% had been reached, platelet recovery was observed regardless of severity [92].

The increased risk of adverse outcomes with secondary dengue infection is well documented. Indeed, there is a correlation between the serologic subtype of dengue and various haematologic parameters [38]. Recent studies have also demonstrated that patients who are triple-positive for IgG, IgM and NS1 were more likely to develop severe-dengue-related thrombocytopaenia compared to those who were only NS1-positive [40]. Given the promising data supporting immuno-haematologic indices as the predictors of outcome in dengue, future studies should evaluate the role of combining clinical and laboratory parameters to create composite predictive scores.

## 6. Implications for Management

There is no licensed anti-viral treatment for dengue infection. The mainstay of management remains supportive care, including management of intravascular volume and bleeding [6]. Transfusions of platelets, fresh frozen plasma and cryoprecipitate can be life-saving in selected patients with severe bleeding. The indication for prophylactic platelet transfusions in dengue infection, however, remains uncertain and is not recommended at present [94]. The normal response to platelet transfusion is an increase in platelet count at 10 min–1 h following transfusion [95]. However, as dengue-mediated thrombocytopaenia is at least partly immune-mediated [91], the transfused platelets are likely to be destroyed at an accelerated rate, which leads to a suboptimal platelet increment.

It has been hypothesised that patients with severe thrombocytopaenia (i.e., platelet count less than 10 × 10^9^/L) are more likely to have immune-mediated platelet destruction and are less likely to benefit from platelet transfusion [91]. A single-centre randomised controlled trial (RCT) demonstrated that, although platelet transfusions increased platelet counts, they had no impact on bleeding-related outcomes and were associated with significant adverse effects [96]. Given the potential side effects of transfusions and their questionable benefit, therapies targeting the immune dysregulation in dengue as a means to treat dengue-related thrombocytopaenia are likely to be a more worthwhile avenue for future research.

Steroids have been used for many years to address the dysregulated host immune response, which causes much morbidity in dengue. However, the evidence supporting this practice is weak. Although non-randomised trials of corticosteroids given for severe dengue showed benefit, there is insufficient evidence to justify the routine use of steroids [97]. A RCT in Vietnam comparing high-dose (2 mg/kg) and low-dose (0.5 mg/kg) prednisolone for 3 days in early dengue infection showed no reduction in the incidence of shock or other complications [98]. Both low-dose and high-dose dexamethasone was trialled in patients with dengue fever with thrombocytopaenia in a separate study, where the control group demonstrated a higher rise in mean platelet counts on all 4 days of assessment, although bleeding manifestations were not significantly improved [99]. An important concern raised with regard to steroid use in dengue is the potential for immunosuppression leading to increased viral replication [100]. These findings were backed up by a recent systemic review, which showed that there was insufficient evidence to justify the use of corticosteroids in DHF and DSS [101]. Larger RCTs are required to evaluate the role of steroids in dengue, given that there may be certain subgroups of patients who may benefit. These trials should incorporate translational correlatives to identify the biomarkers, which may be used as predictors of response to steroids.

It is noteworthy that conventional anti-viral therapies have not shown significant clinical benefit in dengue infection, with balapiravir, and celgosivir [102], among others, having been evaluated in clinical trials. Given these findings, the repurposing of anti-parasitic agents against dengue has been actively explored by several groups. Ivermectin inhibits DENV replication in vitro by targeting the host nuclear import proteins that are required for nuclear localisation of the dengue NS5 protein [103]. A phase 3 RCT demonstrated faster clearance of NS1 antigenaemia in the ivermectin-treated arm; however, there were no clear improvements in viraemia or clinical outcomes [104]. The anti-helminth agent doxycycline was also shown to reduce viral replication in vitro by inhibiting NS2B-NS2 protease activity [105]. Indeed, a RCT of 231 patients demonstrated reduced mortality in the doxycycline-treated arm compared to dengue patients treated with standard supportive care, suggesting the potential of doxycycline as a dengue therapeutic. In addition to the repurposing of older drugs, several novel compounds targeting the DENV are in development [103]. JNJ-A07, for example, was identified via a cell-based screen and inhibited the interaction between the NS4B and NS3 proteins [106,107]. The impact of these novel agents on the immune profile of dengue patients, as well as the role of host immunity in the clinical responses, should be studied in future clinical trials.

## 7. Conclusions and Future Directions

Our understanding of the immuno-haematologic manifestations of dengue infection has seen significant growth in the recent past. While the host immune response plays a pivotal role in the pathogenesis and progression of dengue, our understanding of the underlying mechanisms remains incomplete. A clear delineation of immune cell subsets driving severe dengue at each stage of disease, as well as the corresponding cytokine profile, would be of critical importance. A schematic representation of the proposed role played by various immune cell subsets at each stage in the development of severe dengue is shown in Figure 4. While this provides a framework to understand the immunopathology of severe dengue, further studies are required to gain a deeper understanding of the pathophysiology and to identify therapeutic targets. The role of the host gut microbiome in the progression of dengue is also an active area of study, and while the interaction between gut flora and the host immune response is becoming well established [108], its impact on haematologic manifestations, including bleeding, remains unknown. Given that the gut microbiome is amenable to therapeutic intervention, this would be a key area for future investigations.

Further improvements in patient outcomes will require early prediction of severe disease, as well as novel therapeutics targeting the immune response. Future trials of dengue viral therapeutics and vaccines should include correlative translational research, including high-dimensional immune profiling analyses. This will provide an integrated picture of the host immune system in relation to haematologic parameters in dengue infection and inform how best they can be modulated to improve clinical outcomes.

## Figures and Tables

**Figure 1 viruses-16-01090-f001:**
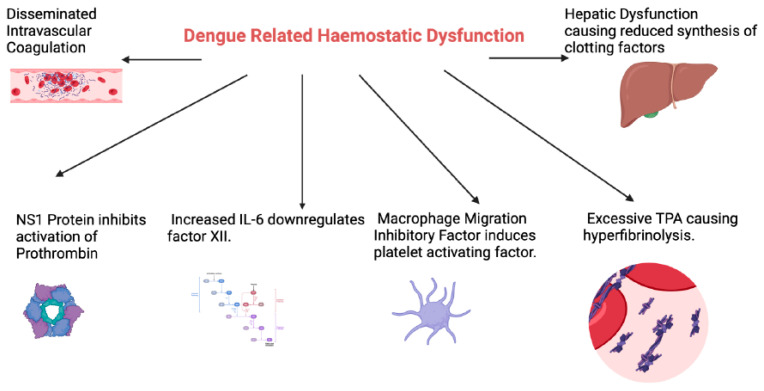
An overview of the mechanisms underlying dengue related haemostatic derangements. NS1: Dengue non-structural antigen 1; IL-6: Interleukin 6; TPA: Tissue plasminogen activator. (Figure created with Biorender.com).

**Figure 2 viruses-16-01090-f002:**
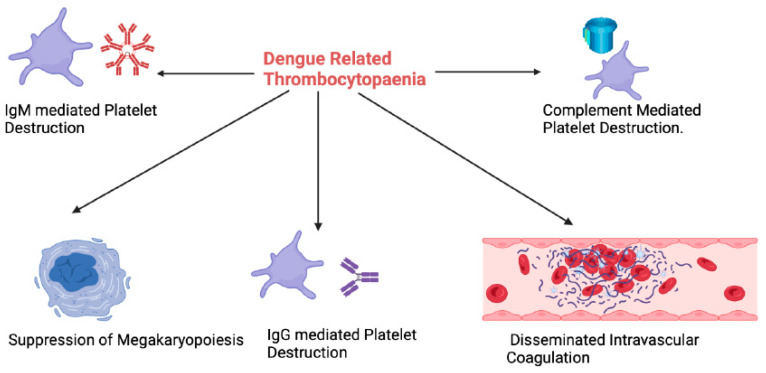
An overview of the mechanisms behind dengue related thrombocytopaenia. IgM: Immunoglobulin M; IgG: Immunoglobulin G. (Figure created with Biorender.com).

**Figure 3 viruses-16-01090-f003:**
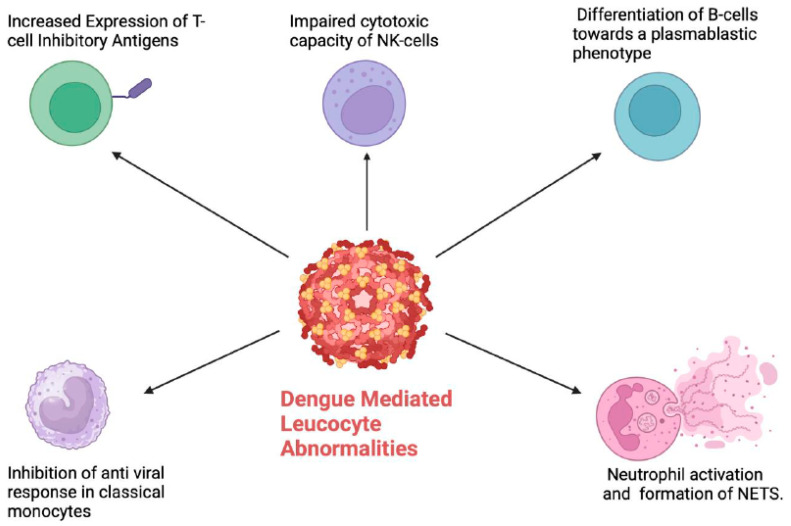
Dengue mediated leucocyte abnormalities resulting in a dysregulated host immune response. NETS: Neutrophil extracellular traps. (Figure created with Biorender.com).

**Figure 4 viruses-16-01090-f004:**
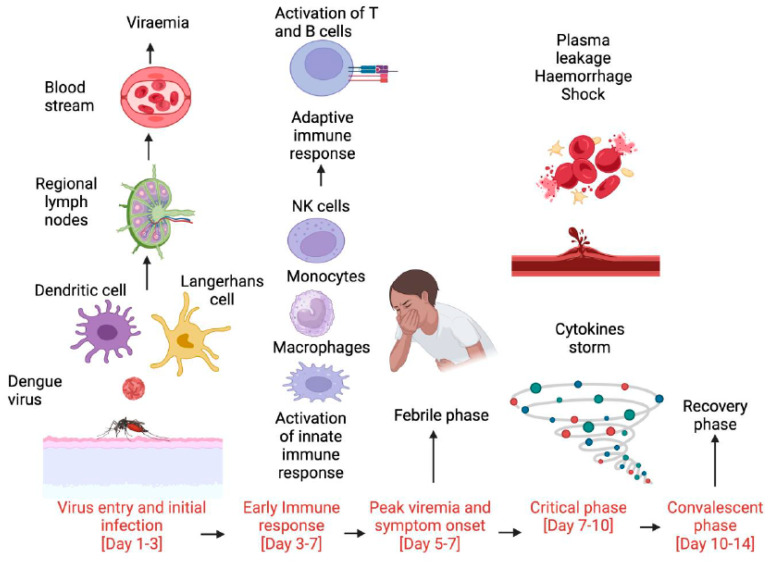
An overview of the roles of immune cell subsets at each stage in the pathogenesis of severe dengue. (Figure created with Biorender.com).

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
