# Peer review of "Immuno-Haematologic Aspects of Dengue Infection: Biologic Insights and Clinical Implications"

_viruses, 2024, doi:10.3390/v16071090_

Round 1

Reviewer 1 Report

Comments and Suggestions for Authors

This review comprehensively described the immuno-hematological aspects of dengue infection including hemostatic abnormalities, thrombocytopenia, leucocyte abnormalities (containing different subsets such as monocyte, lymphocyte, and neutrophil), and further applications these findings into diagnosis and management of dengue. However, I am interested in some part of the description.

1.     The authors mentioned that elevated levels of vWF antigen, tissue factor, and PAI-1 were observed in children with DHF. All these factors are typically associated with coagulation and contribute to thrombus formation. Could the authors explain why the elevated level of these coagulation factors causing hemorrhage in DHF? Are similar findings observed in adult patients with dengue virus infection?

2.     Excessive vWF and platelet binding leading to enhanced platelet clearance, as well as abnormal platelet activation and aggregation upon dengue virus infection may contribute to thrombocytopenia and bleeding (References: doi: 10.1371/journal.ppat.1007500, doi: 10.1371/journal.ppat.1007625).

3.     In addition to inhibiting activation of prothrombin, NS1 mediates activation of plasminogen, thereby enhancing fibrinolytic activity (References: doi: 10.4049/jimmunol.1500057. Epub 2015 Dec 28.). With these references, perhaps the authors could further elaborate pathophysiological mechanism of dengue induced thrombocytopenia and coagulopathy.

Suggested corrections:

1.     Abstract: “Aedes aegypti” and “Aedes albopictus” need to be italicized.

2.     Figure 1: Hepatic dysfunction causing reduced synthesis o”f” clotting factors….lack of an “f”.

3.     Page 4 line 92: the reference 18 may be miscited. This reference didn’t mention the effect of NS1 in inhibiting the activation of prothrombin.

4.     Page 6 line 236: the Fcy receptors FcyRIIB …It’s Fcγ, not y.

5.     Page 7 line 245: the same problem, IFN-y production…It’s “γ”, not y.

6.     Page 4 line 129: plasma levels of platelet factor-4 and beta-thrombomodulin are increased…according to the reference 32, I think it might be beta-thromboglobulin not thrombomodulin.    

7.     Page 9 line 344: platelet count less than 10 x 10”^”9/L)

Author Response

Please see the attachment. (Point-by-point response to reviewer one comments)

Reviewer 2 Report

Comments and Suggestions for Authors

The author has thoroughly reviewed the possible immuno-hematologic mechanisms of pathogenesis following dengue virus infection in this manuscript. However, I have the following suggestions:

1.      Page 2 Line 59: NS-1à NS1 for consistency

2.      Page 2 Line 67: Figure1à Figure 1

3.      I suggest the author cite Figure 1 in the section “Haemostatic Abnormalities in Dengue Infection”.

4.      The author should describe the correlation between hepatic dysfunction and reduced synthesis of clotting factors as mentioned in Figure 1.

5.      The author should choose between the spellings "Thrombocytopaenia" and "Thrombocytopenia" and use the selected spelling consistently throughout the manuscript.

6.      The author mentioned that a specific subset of lymphocytes, rather than total lymphocytes, may be more relevant in the pathophysiology of dengue infection. Which subset of lymphocytes was referred to in their articles? Please provide more information about this phenomenon and discuss the possible mechanisms, as this would likely interest readers.

7.      Splenic macrophages and monocytes have been suggested as target cells for dengue virus infection among blood mononuclear cells (PMID: 17928355 and 18041019). I suggest the authors expand the discussion on this topic.

8.      Since the pathophysiological mechanism in a murine model is not always applicable to humans, I suggest the authors remove the content related to the murine model. If the authors wish to retain the content about mice, I suggest creating a separate section to explain the dengue pathogenesis in mice, which will help prevent readers from confusing information from the two different systems.

9.      Some studies have shown that age and chronic diseases are related to the severity of dengue. I suggest the authors consider including these results in the section "Implications for Prognosis and Risk Stratification".

10.  Page 8 Line 314: reference 9596 should be corrected.

11.  As the author mentioned, different immune cells are involved in each stage of dengue virus pathogenesis. I am curious if the author could create a diagram to illustrate the timeline of viral and immunopathological mechanisms in dengue virus pathogenesis.

Comments on the Quality of English Language

Minor editing of the English language is required.

Author Response

Please see the attachment.(Point-by-point response to reviewer two comments.)

Reviewer 3 Report

Comments and Suggestions for Authors

Authors reviewed the landscape of dengue virus infection and disease and helped researcher to understand Immuno-haematologic Aspects of Dengue Infection: Biological Insights and Clinical Implications.

Antibody-dependent enhancement (ADE) of dengue is an immunologic aspect of disease that is very important exacerbated dengue.   Authors discuss different immune cells and populations and how they modulate clinical dengue outcomes. They failed to dwell extensively on B-cells in the context of antibodies and their contribution to disease, especially in areas where dengue serotypes or flaviviruses co-circulate. It would be appropriate to shed more light on ADE of denque virus infection and diseases to give researcher an overall view of the biologic implications of the disease. 

Author Response

Please see the attachment.(Point-by-point response to reviewer three comments.)

Round 2

Reviewer 2 Report

Comments and Suggestions for Authors

The author has significantly improved the manuscript; however, several major errors have emerged in the revised manuscript. It is recommended that the author carefully revise it to meet the fundamental requirements of a high-standard journal.

1.      There are two articles listed under reference 55 in the bibliography; please make the necessary corrections.

2.      The content of reference 100 does not mention dengue.

3.      The textual description of reference 103 in the "Implications for Prognosis and Risk Stratification" section does not seem to align with the content of the reference.
